# Overview on Common Genes Involved in the Onset of Glioma and on the Role of Migraine as Risk Factor: Predictive Biomarkers or Therapeutic Targets?

**DOI:** 10.3390/jpm12121969

**Published:** 2022-11-28

**Authors:** Giovanna Casili, Marika Lanza, Alessia Filippone, Maria Caffo, Irene Paterniti, Michela Campolo, Lorenzo Colarossi, Dorotea Sciacca, Sofia Paola Lombardo, Salvatore Cuzzocrea, Emanuela Esposito

**Affiliations:** 1Department of Chemical, Biological, Pharmaceutical and Environmental Sciences, University of Messina, 98122 Messina, Italy; 2Unit of Neurosurgery, Department of Biomedical and Dental Sciences and Morphofunctional Imaging, University of Messina, 98122 Messina, Italy; 3Istituto Oncologico del Mediterraneo, Via Penninazzo 7, 95029 Catania, Italy

**Keywords:** glioma, CLOCK, BMLA1, NOTCH, diagnostic, therapeutic targets

## Abstract

**Simple Summary:**

Mutations in hundreds of genes have been identified in gliomas, and most relevant discoveries showed specific gene alterations as potential risk factors for brain tumor onset. The aim of this study was to clarify the function of these genes in triggering or modulating brain tumors, to early diagnosing brain tumor onset in patients affected by a simple headache. We confirmed a significant modulation of CLOCK, BMLA1 and NOTCH genes in glioma, particularly praising NOTCH genes family to be good as potentially attractive therapeutic targets for glioblastoma, strengthening the protective role observed in clinical trials for brain tumors. The investigation of these genes could suggest potential therapeutic targets for treating brain tumors, open up the possibility of personalized treatments that can target each brain tumor’s specific genetic abnormality.

**Abstract:**

Gliomas are relatively rare but fatal cancers, and there has been insufficient research to specifically evaluate the role of headache as a risk factor. Nowadays, gliomas are difficult to cure due to the infiltrative nature and the absence of specific adjuvant therapies. Until now, mutations in hundreds of genes have been identified in gliomas and most relevant discoveries showed specific genes alterations related to migraine as potential risk factors for brain tumor onset. Prognostic biomarkers are required at the time of diagnosis to better adapt therapies for cancer patients. In this review, we aimed to highlight the significant modulation of CLOCK, BMLA1 and NOTCH genes in glioma onset and development, praising these genes to be good as potentially attractive therapeutic markers for brain tumors. A improved knowledge regarding the role of these genes in triggering or modulating glioma maybe the key to early diagnosing brain tumor onset in patients affected by a simple headache. In addition, investigating on these genes we can suggest potential therapeutic targets for treating brain tumors. These considerations open up the possibility of personalized treatments that can target each brain tumor’s specific genetic abnormality.

## 1. Introduction

Gliomas are relatively rare but fatal cancers [1]. The symptomatology of brain tumor patients appears complex, although a classical sign is represented by headaches. Various studies have described the features of headache as a symptom of brain tumors; particularly, headaches more often occur in approximately 33–71% of brain tumor patients, although there is a lack of data on the risk and association of headache on the development of brain tumors [2,3]. Although very few studies have attempted to specifically evaluate the role of headaches as a risk factor, there is no correlation between the clinical phenotype of headache and brain tumor characteristics and [4].

Glioma represents the most aggressive brain tumors, and among these, glioblastomas (GBMs) are responsible for significant morbidity and mortality in both pediatric and adult populations. Current therapies include surgical resection, followed by radiotherapy plus simultaneous treatment and maintenance with temozolomide (TMZ) [4]. The standard Stupp regimen for GBM patients, which was subsequently recommended as the international consensus guideline for GBM treatment, consisting in fractionated focal irradiation plus concomitant daily TMZ does not guarantee a remission of the disease, presenting GBM as an important therapeutic challenge due to their poor outcome [5]. Unfortunately, standard therapies are often ineffective, and average survival time for GBM patients is not more than 18 months [6]. Therefore, the development of methods to prevent the onset of brain cancer or to detect brain tumors at an early stage is extremely relevant [1].

Several important genetic alterations have been identified in brain tumors, and the most relevant findings have shown specific gene mutations. Interestingly, new technologies have allowed much deep genetic and epigenetic analysis of large numbers of glioma samples, leading to a number of novel discoveries. One of the most exciting and clinically relevant observations was the discovery that a high percentage of glioma and a very small percentage of primary glioblastoma harbor mutations in the isocitrate dehydrogenase 1 (IDH1) gene. Moreover, although glioma-specific mutations are seen, mutations in common cancer genes, such as TP53 and PTEN, are very frequent, but are not of prognostic importance [7].

Predictive biomarkers are needed at the time of or before diagnosis to better tailor therapies for cancer patients. Therefore, it remains interesting to investigate which genes overlap in brain tumorigenesis.

Particularly, a significant correlation between CLOCK, BMLA1, and NOTCH genes, involved in brain tumor progression, appear to emphasize the risk in patients with migraine to develop brain tumor. The expression of the CLOCK gene is augmented in ~5% of GBM patients and high-grade gliomas compared to low-grade gliomas or control. Scientific data suggest the tumor-progressing role of CLOCK gene in gliomas, indicating that CLOCK could stimulate the proliferation and migration of glioma cells through the inflammatory NF-κB signaling pathway. In addition, CLOCK stimulates tumor spread in different brain tumor models and regulates cancer metabolism. Considering this evidence, targeting the circadian clock by regulating the CLOCK could be a promising approach to treat glioma [8].

In addition, glioblastoma cell lines (GSCs) derived from patient and human GBM cell cultures showed circadian rhythms in brain and muscle Arnt-like protein 1 (Bmal1) expression. Numerous studies have suggested that GBM cells show a purposeful clock regulating many cellular pathways, including oxidative stress and other metabolic and energetic activities. Recently, the involvement of cellular clocks in tumor development and cellular survival got stronger. In gliomas, upregulation or downregulation of BMAL1 expression is also significant. An upregulation in BMAL1 expression was reported in high-grade glioma patients, highlighting that BMAL1 may work as a tumor suppressor. Recent discoveries highlighted a correlation between BMAL1 overexpression and a decrease in glioma invasiveness, through the inhibition of PI3K/AKT/matrix metalloproteinase-2 signaling pathway [9]. Accordingly, mutations in NOTCH genes could provoke up- or downregulation in NOTCH signaling, as has been observed in several types of cancer [10]. The aim of this review was to investigate the most common genes, CLOCK, BMLA1 and NOTCH, involved in the development of gliomas, as potential diagnostic or therapeutic targets to prevent or reduce the occurrence of brain cancer. The genetic overlap analysis—given that genes are the predominant functional unit of the human genome and more closely related to biology—could provide novel evidence on the genetic profile of glioma, give insight into shared biological pathways underlying the disorder and help identify target genes as predictive markers or therapeutical target. Therefore, a more thoughtful approach to the impact of circadian clock genes and NOTCH genes on some mechanisms of the tumors crossways the glioma setting could be crucial to discover new therapeutic strategies and to translate these findings into the clinical setting.

## 2. New Insight on Molecular Involvements of Gliomas

The incidence of brain tumor is about 6 per 100,000 persons per year [11]. Specifically, glioma strikes people over 55 years of age, with a higher incidence in men than in women [12]. Although GBMs account for only 15% of primary brain cancer, it represents the most malignant form due to its poor prognosis.

Infiltrative growth, intra-tumoral heterogeneity and tumor reversion are all characteristics of glioma. The World Health Organization (WHO) classification lists primary brain tumors on histopathologic, immunohistochemical and biomolecular data and malignancy grade [13,14].

The discovery of molecular mechanisms regarding radio/chemo resistance and biomarker detection could improve the strategy for novel therapeutics [15,16,17].

Symptomatologically, gliomas are characterized by persistent headaches, fatigue, general discomfort, behavioral changes, such as fluctuations in personality, weakness or paralysis or vision or speech disorders [18,19]. The therapeutic approaches commonly used are surgery, radiation and chemotherapy, although they are ineffective for glioma progression. Clinically, advantages of maximal resection include relief of mass effect, decreased tumor burden and prolonged survival [20]. Radiotherapy can be used as adjuvant therapy following surgical resection; other options include stereotactic radiosurgery, usually for older or selected patients [21,22]. In addition, chemotherapy given in combination with radiation has been shown to improve survival in patients with high-grade gliomas [23].

Chemotherapy represents the typical approach following surgery sideways with alongside radiotherapy; however, overcoming blood-brain barrier (BBB), interaction with anti-epileptic drugs and intrinsic or acquired resistance represent the restricting factors for chemotherapy. Mutations of various genes, such as TP53, PTEN, CDKN2A and EGFR, were found to be frequent in glioma tumorigenesis [12].

So, the relationship between genetic and epigenetic events suggests a mechanism behind the histone mutation. Unconventional enlargement of telomeres and explicit gene expression profiles associated with H3F3A/ATRX-DAXX/TP53 mutations have also been described. The literature reports somatic mutations in over 40% of tumor events and recurrent mutations in over 30% of tumors, showing it as mutation targets on the histone tail. Consequently, the pharmacological inhibition of histone demethylation could support glioma management. The discovery of isocitrate dehydrogenase (IDH) genes in the pathogenesis of glioma has been documented as a prognostic marker in high grade-gliomas and GBMs. Specifically, IDH1 or IDH2 are usually mutated in WHO grade II/III glioma or secondary GBM patients, mainly in the frontal lobe (75%) and temporal lobe (40%). Therefore, the valuation of IDH mutation is of significant diagnostic relevance and therapeutic value. Inhibitors of IDH mutant proteins might also serve in glioma stratification [14,24]. The targeted approaches currently in clinical trials or in laboratory development include drugs, monoclonal antibodies, immunotherapy, small molecules inhibiting specific proteins and specific targeting of cancer stem cells (CSCs). Gene therapy disclosed remarkably helpful efficacy in pre-clinical phase, but it failed in phase III trials because of the heterogeneity and invasiveness of gliomas [25]. At the same time, immunotherapeutic approaches focus on the enhancement of T-cell function by generally stimulating the immune system or by attacking specific tumor cell antigens. These approaches include the use of vaccines, adoptive cell transfer and immune checkpoint inhibitors.

A hallmark feature of gliomas is an abundance of infiltrating immune cells, wherein microglia are known to contribute to an immunosuppressive microenvironment and support glioma progression, and CLOCK-regulated microglia recruitment are consistent that the immune system can be synchronized by circadian components. Recent studies have suggested that CLOCK has a tumor-promoting function in gliomas [26]. Notably, CLOCK might increase the proliferation and migration of glioma cells through the NF-κB signaling pathway [27]. Differently, Wang et al. (2021) documented that CLOCK is downregulated in GBM samples [28]. Taking into consideration this proof, targeting the circadian clock by altering the CLOCK function could be a promising strategy for glioma and GBM treatment (Figure 1).

Circadian clock genes regulate several positive and negative feedback loops and generate circadian oscillations [29]. Core clocks include various genes, including BMAL1 [30]; all genes belonging to the CLOCK family are located in the suprachiasmatic nucleus (SCN) of the hypothalamus and in all peripheral tissue cells [31]. CLOCK genes, in addition to regulating their own circadian rhythm, could control the expression of about 40% of protein-coding genes in the genomes of mammals, suggesting that numerous important and complex physiological functions can be regulated through its rhythmic expression [32]. Therefore, the abnormal expression of clock genes and disruption of circadian rhythms can be strikingly involved in the progress of several diseases, including tumor pathology. Animal and human studies have shown that debilitating mutations or disruptions of core circadian genes can have adverse effects on metabolic, physiologic and neurologic processes [33].

### 2.1. CLOCK Gene

CLOCK is a gene, located at the 4q12 chromosomal region, encoding transcription factor which is believed to affect both the persistence and period of circadian rhythms. Research shows that the CLOCK gene plays a major role as an activator of downstream elements in the pathway critical to the generation of circadian rhythms [34,35,36,37]. Interestingly, two single-nucleotide polymorphisms (SNPs, rs12649507 and rs11932595) located in the intronic region of the CLOCK gene were associated with sleep duration. Brain tumors display circadian behavior [38], and the disruption of the circadian clock is associated with a higher incidence of tumors [39]. In some tissues, the CLOCK gene can be designated to promote cancer, acting as an oncogene, whereas in others, it acts as a tumor inhibitor. The origin of such differences is an open question that, when answered, will help investigators to identify the mechanisms by which tumor cells capture the molecular clock machinery and manages it in his favor [40]. Interestingly, two key CLOCK genes act as oncogenes in glioma; particularly, genes are essential for the survival and proliferation GSCs in vitro. By contrast, neither differentiated GBM cells or normal neural stem cells seem to be contingent on the genes in this way, and the theory was corroborated by showing a strong correlation between the expression of some of the core clock components and patient outcomes [41]. The circadian clock output of GSCs includes genes convoluted in the metabolic pathway (glucidic and lipidic), whereas the circadian clock output of normal neural stem cells does not; therefore, variations in glucose metabolism promote cancer progression [42], probably affecting fatty acid metabolism and glycolysis, to increase the energy necessary for the high rate of proliferation of cancer cells [43]. Due to their rapid growth, neuronal cancer cells display an increased requirement for nutrients, shifting their metabolism toward the utilization of glycolysis rather than oxidative phosphorylation to generate ATP [44]; thus, various metabolic pathway components are altered in tumors.

The disruption of the biological clock by environmental factors or through mutations in the circadian pathway can lead to an increased risk of tumorigenesis [45,46,47]. Discrepancies in the expression of the CLOCK gene in cancer provokes alterations in the activation and/or inhibition of the main oncogenic and tumor suppressive pathways [48]. Recently, it has been demonstrated that after clock inhibition, a reduction in proliferation and induction of apoptosis was observed in glioma cells, related to an upregulation of p53 complex, emphasizing the anti-apoptotic inflection of CLOCK in gliomas [6].

A new study described an inverse correlation between the tumor suppressor activity of CLOCK and hypoxia [49], supporting the thesis that the suppression of CLOCK in tumor cells enhanced survival and reduced migration of the microglia [50]. Future works should increase the acknowledgement of clock genes in glioma research [51]. The knowledge of the CLOCK mechanism in tumors could be important for the progress of tumor therapy. Additionally, TGF-β acts as an important regulator of the physiological clock, that regulates the expression of positive and negative switches of circadian rhythm oscillation [52].

### 2.2. BMAL1 Gene

Among the many different CLOCK genes, BMAL1 is the only gene whose removal can abolish circadian rhythmicity [53]. BMAL1 is a key element of the cell-autonomous transcription translation feedback loops [54]. Additionally, circadian clock modification due to BMAL1 mutations accelerates tumor growth or the whole carcinogenesis process [55]. This is due to a growth proliferation rate upon circadian rhythm disruption because tumor suppressor and key cell cycle genes are under CLOCK control. In support of the relationship between rhythm disruption and oncogenesis, deregulated circadian rhythms appear to be a common feature of cancer cell lines and advanced-stage tumors [56,57]. Knocking down the essential clock gene BMAL1 in B16 tumors prevented the effects of dexamethasone on tumor growth and cell cycle events [58].

The expression of BMAL1 is involved in glioma biology, both when it is upregulated and downregulated. Interestingly, BMAL1 overexpression in high-grade glioma patients promoted it as a tumor suppressor in GBM cell growth.

### 2.3. CLOCK/BMAL1 Gene as Therapeutic Target

The identification of functional molecule regulators of individual clock proteins that may not be necessarily linked to their circadian function could offer a more specific therapeutic drug. Various advances have been recently performed in a screening for modulators of CLOCK/BMAL1, highlighting that different types of circadian mutants displayed an opposite response to toxicity induced by chemotherapeutic agents [59,60]. Another example of the therapeutic value of drugs targeting CLOCK/BMAL1 functional activity came from a series of in vivo studies in Cry-deficient mice. It has been found that disruption of the circadian clock by a mutation in p53-null background makes them more sensitive to apoptosis [61]. Mechanistically, upregulation of p73, a member of the p53 family, in the absence of Cry, correlates with increased levels of the early growth response 1 (Egr1) gene, which works as a positive activator of p73 and which itself is directly regulated by the CLOCK/BMAL1 transcriptional complex [62]. Interestingly, CLOCK promotes microglia infiltration in gliomas. Microglia are well known to be immunosuppressive cells in brain tumors and may therefore reduce immune checkpoint blockade activity. Particularly, GBM are highly infiltrated with tumor-associated microglia and macrophages (TAMs), which are known to support GBM cells and promote tumor progression. Several factors are involved in the crosstalk between GBM cells and TAMs, most of which attract and recruit TAMs to the tumor or polarize the TAMs toward a more pro-tumorigenic M2-like phenotype; thus, the tumor modifies and exploits the TAMs to support its malignant progression [63].

Additionally, demonstrating the capacity of CLOCK to specifically and directly regulate chemokines, such as OLFML3, novel chemokine recruiting immune-suppressive microglia into the tumor microenvironment, which in turn recruits microglia into the GBM, encourages the design of clinical trials targeting OLFML3 in high CLOCK GBM patients and abundant microglia [62]. Thus, it is tempting to speculate that combined inhibition of OLFML3 and immune checkpoint blockade may also prove beneficial for GBM patients [64].

Novel concepts in glioma therapy is chronotherapy, described as a therapeutic approach based on patients’ circadian rhythms. The goal of this treatment is to determine the optimal time of the day to assess therapy to better outline outcomes with the most efficient drug doses, reducing drug toxicity and side effects. Tumor tissues exert varying biological activities compared to normal tissues due to resetting of altered rhythms; thus, chronotherapeutic compounds used for cancer treatment should exploit the timing of circadian rhythms to achieve higher efficacy and mild toxicity. Commonly, chronotherapies depend on the circadian timing system that controls circadian rhythms involving metabolism and biological activities. In particular, chronotherapy aims to increase antitumor effects and to minimize the toxicity of anticancer agents in normal tissues. Patients who received chrono-modulated infusions of 5-FU, leucovorin and oxaliplatin at separate times showed a low frequency of side effects [65,66]. Accumulating evidence reveals that the circadian rhythm has an impact on glioma treatment; particularly, a recent study highlighted that GBM cells exhibited a significant temporal response to bortezomib, used as a proteosome inhibitor in recurrences stages of GBM treatment [67,68]. Moreover, Wagner et al. [17] demonstrated the higher efficacy of low-dose Bortezomib treatment when administered in tumor-bearing animals at night compared to day/night administration, although the effect of chronotherapy treatment in glioma is still controversial and needs further large-population-based trials for validation.

## 3. NOTCH Gene

NOTCH proteins are a family of type-1 transmembrane proteins, forming a core component of the signaling pathway. NOTCH proteins appear to act as transmembrane receptors for intercellular signals; the known are: NOTCH1, NOTCH2, NOTCH3 and NOTCH4 [69].

NOTCH signal activation depends on direct interaction of one of the four NOTCH receptors (NOTCH1–4); particularly, ligand-induced proteolysis of NOTCH receptors release the NOTCH intracellular domain that control the expression of a wide range of specific target genes, suggesting its multiple functions, including cell proliferation, stem cell maintenance, cell fate decisions and differentiation [70].

The NOTCH signaling network is an evolutionarily conserved intercellular signaling pathway that regulates interactions between physically adjacent cells. It represents a critical regulator of differentiation programs; influencing cell cycle kinetics and apoptotic signals, it can be suggested that NOTCH proteins may be involved in the malignant transformation of selected cell systems. Alterations in NOTCH signaling are visible in various disease [71]. Interestingly, the most common inherited cause of stroke and vascular dementia in adults are represented by a genetic mutation NOTCH3 gene; particularly, cerebral autosomal dominant arteriopathy with subcortical infarcts and leukoencephalopathy (CADASIL) symbolizes the genetic mutation in the neurogenic locus of NOTCH3 gene [72,73].

Numerous evidence shows NOTCH participation in carcinogenesis and human tumors. NOTCH activation has been mainly associated with its multiple effects in sustaining oncogenesis, including tumor cell proliferation, migration, cell cycle progression and apoptosis inhibition [74,75]. Due to its pleiotropic function, the NOTCH signaling pathway is involved in many aspects of tumor development and may act as either an oncogene or a tumor suppressor. The balance between one role and another is determined by many different factors [76]. Indeed, NOTCH signaling is constitutively activated in several types of cancer cells, acting as an antiapoptotic or pro-oncogenic signal. NOTCH3 overexpression has been reported to be responsible for increased in vitro tumor cell growth in human lung cancer and NOTCH3 constitutive activation was reported to inhibit terminal differentiation in lungs of transgenic mice. Moreover, increased expression of NOTCH3 has been observed in human tumors and also participates in the induction of terminal differentiation and growth arrest [77]. Excitingly, different mutation clusters within the NOTCH1 receptor gene significantly encourage tumor progression by causing a ligand-independent constitutive activation of the pathway [78]. Brain cancer data in The Cancer Genome Atlas (TCGA) (https://www.cancer.gov/) were downloaded from cBioPortal for cancer genomics, including gene expression profiles of NOTCH genes and clinical information, observing an increasing variance incidence predominantly for NOTCH1. Notably, NOTCH signaling hypothetically controls multiple steps of gliomagenesis, comprising tumor initiation, progression and recurrence, although a clarified involvement of NOTCH in glioma development is still missing. Furthermore, NOTCH signaling plays a central role in maintaining the quiescent NSC pool [79], which result to be resistant to treatment and can regenerate proliferating progenitor cells. As well, NOTCH signaling activity can increase stem cell features, supporting resistance to radio- and chemo-therapies and activating oncogenic pathways or inhibiting tumor suppressor. On the other hand, NOTCH-inactivating mutations and low expression levels of canonical NOTCH target genes have been documented in patients with glioma, indicating a tumor-suppressive role of NOTCH. The data obtained using cBioPortal for cancer genomics allowed us to analyze the frequency of mutations in NOTCH genes, providing e a model of gene interaction to be studied in depth in the pathogenesis of brain cancer (Figure 2). Among various epigenetic alterations, such as acetylation, phosphorylation, ubiquitylation and sumoylation, promoter region methylation is considered as an important component in cancer development. Various studies have shown that DNA methylation is reversible, and demethylating drugs can reverse the silencing of genes resulting from methylation. Regarding NOTCH genes, not much is known about the methylation status in cancer. Interestingly, brain cancer data in The Cancer Genome Atlas (TCGA) (https://www.cancer.gov/) downloaded from cBioPortal for cancer genomics, highlighted an increasing methylation status (HM450) predominantly for NOTCH3, compared to other NOTCH genes, as shown in Figure 3. Deletions or shallow deletions were corresponding to lower mRNA expression, while copy number gains or amplifications were linked to increased mRNA expression of the NOTCH genes. The shallow deletions of NOTCH1 and NOTCH2 are the most frequent copy number loss of these NOTCH genes, while the gain of NOTCH3 and NOTCH4 DNA copy number represents the most common change in gene amplification (Figure 3). A better confirmation of the role of glioma in the involvement of the NOTCH gen, is shown by analyzing the NOTCH pathway in glioma downloaded from cBioPortal for cancer genomics, highlighting that 7.4% of glioma patients show a mutation in NOTCH1 gene (Figure 4). Interestingly, a combined study on NOTCH gene mutations in glioma and GBM and patient’s survival (data obtained from cBioPortal) highlighted that the patient’s survival was lower in patients with NOTCH4 mutations (11 months) and NOTCH3 (13 months) compared to NOTCH1 (77 months) and NOTCH2 (19 months). Also, patients with double mutations in NOTCH3/NOTCH4 showed a reduced survival (18 months) compared to double mutations in NOTCH1/NOTCH2 (57 months) (Figure 5).

### NOTCH Gene as Therapeutic Target

NOTCH receptors and their targets seem to be likely aspirants for specific drug targeting, and various strategies to control NOTCH are being implemented for cancer therapy [80]. However, consideration of the molecular basis of NOTCH oncogenic and tumor-suppressive functions is indispensable to develop successful strategies to therapeutically check NOTCH in glioma [81,82]. A combined approach of tumor explants with NOTCH inhibition was found to be more effective than either treatment alone, proposing a synergistic effect. Given its significant role in tumor biology, various inhibitors, including gamma secretase inhibitors (GSIs) and targeted monoclonal antibodies, have received increasing attention during the past decade in cancer treatment. The use of GSIs for cancers is primarily based on the premise that GSIs act by inhibiting the cleavage of γ-secretase, which result in blocking NOTCH1 signaling; modulation of the NOTCH pathway increases glioma cell sensitivity to temozolomide [83]. Between these, RO4929097 is an orally bioavailable GSI, a potent inhibitor of Notch signaling, assessed in early phase trials in solid tumors, alone or in combination with other agents, with responses observed in a range of brain tumor types [84].

## 4. Future Prospective

The analysis of circadian role could lead to the identification of prognostic markers and therapeutic targets for cancer, contributing to the development of personalized medicine. The same approach is applicable for all components of the NOTCH signaling pathway; so, targeted therapeutic tools were developed in their molecular structure and post-translational changes [85]. These discoveries praise NOTCH genes as potentially attractive therapeutic targets for brain tumors, strengthening the protective role observed in clinical trials for brain tumors. Considering the tumoral divergency in glioma, it is conceivable that NOTCH inhibition could be useful only in a proportion of molecularly selected patients in a personalized therapy. Whether NOTCH activity in tumor cells can control interactions with the glioma microenvironment and immune shirking needs to be discovered. Therefore, CLOCK and NOTCH represent an interesting approach as predictive markers and therapeutic targets (Figure 6).

## 5. Conclusions

The review offers new insights into the potential role of genetics in brain tumors. A better understanding of the variable etiologies is crucial to appropriate diagnostic evaluation in patients and to propose a correct treatment. Further investigation into the role of CLOCK, BMLA1 and NOTCH genes involved in brain tumor progress, and a better understanding of these genes potential in triggering or modulating brain cancer may prove to be the key to early diagnosis of glioma in patients affected by a modest headache and strengthening the therapeutic role of these genes already known in clinical trials. These considerations open up the possibility of personalized treatments targeting a brain tumor’s specific genetic abnormality.

Signaling pathways involving neurotransmitters could be altered in tumors and can re-entrain the cancer cell clock, thus integrating whole-body information into the cell’s circadian output; these systemic pathways might represent therapeutic targets to treat cancer, although the complex effect of these pathways on circadian reprogramming of cancer cells is still poorly understood. Recent progress in the glioma field, specifically in GBM, support their translational use in patients under clinical trials as a new target for the innovative therapeutic stratagems or to enhance current therapies pointing to decrease tumor growth. A broader and more varied knowledge regarding the impact of CLOCK genes and NOTCH genes on all components of cancer across the glioma landscape and the identification of genetic overlap shared across gliomas could be used to assess the validity of the clinical diagnosis and could be the key to translating the discoveries in a clinical setting.

## Figures and Tables

**Figure 1 jpm-12-01969-f001:**
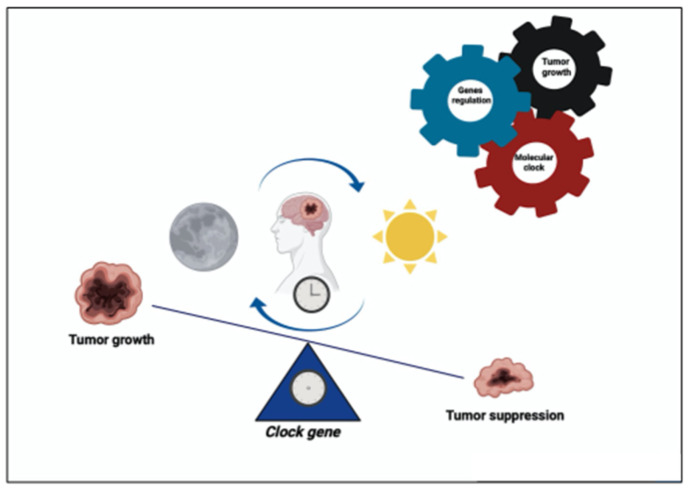
Molecular involvement in gliomas (created by Biorender). Clock genes modulates glioma progression by affecting both tumor immune infiltration and cell proliferation and tumor suppression.

**Figure 2 jpm-12-01969-f002:**
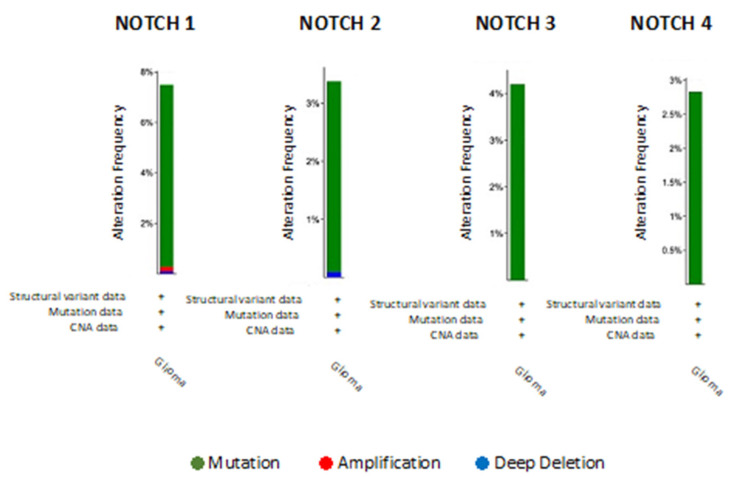
Mutation frequency of NOTCH genes in glioma (data obtained from cBioPortal). The alteration of frequency mutation of NOTCH gene family differs depending on NOTCH genes. The highest frequency rediscovered in glioma concerned the mutation of NOTCH 1 (about 8%), compared to the mutations relative to NOTCH2 (3.6%), NOTCH3 (4.3%) and NOTCH4 (2.8%).

**Figure 3 jpm-12-01969-f003:**
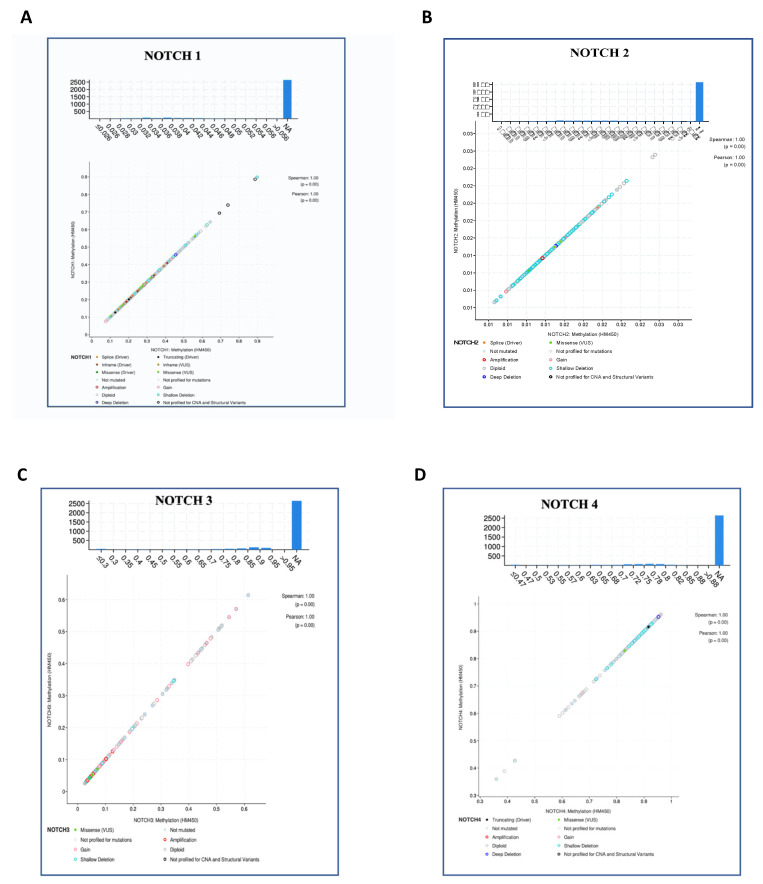
(**A**–**D**). Methylation (HM450) of NOTCH genes in glioma (data obtained from cBioPortal). The data, generated by cBioPortal, refers to CpG imputation ensemble for DNA methylation levels across the human methylation450 (HM450), highlighting the relationship between copy number alterations (not mutated, amplification gain, diploid, shallow deletion) and the expression levels of NOTCH genes.

**Figure 4 jpm-12-01969-f004:**
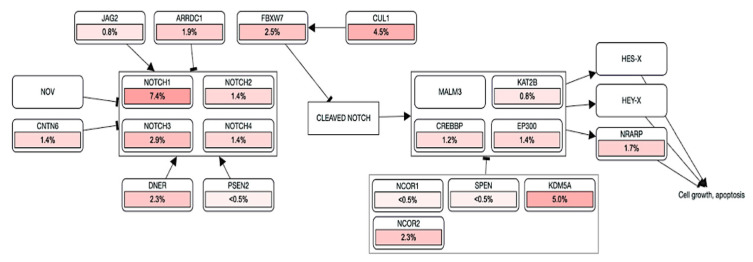
Pathway mapper of NOTCH genes in glioma (data obtained from cBioPortal). The relationship between the genes in the NOTCH signaling pathway. The genes that are partially highlighted and color-coded are significantly associated with the NOTCH pathway in glioma and GBM. The different % displayed the frequency of mutations of various genes in glioma and GBM and the correlation between them.

**Figure 5 jpm-12-01969-f005:**
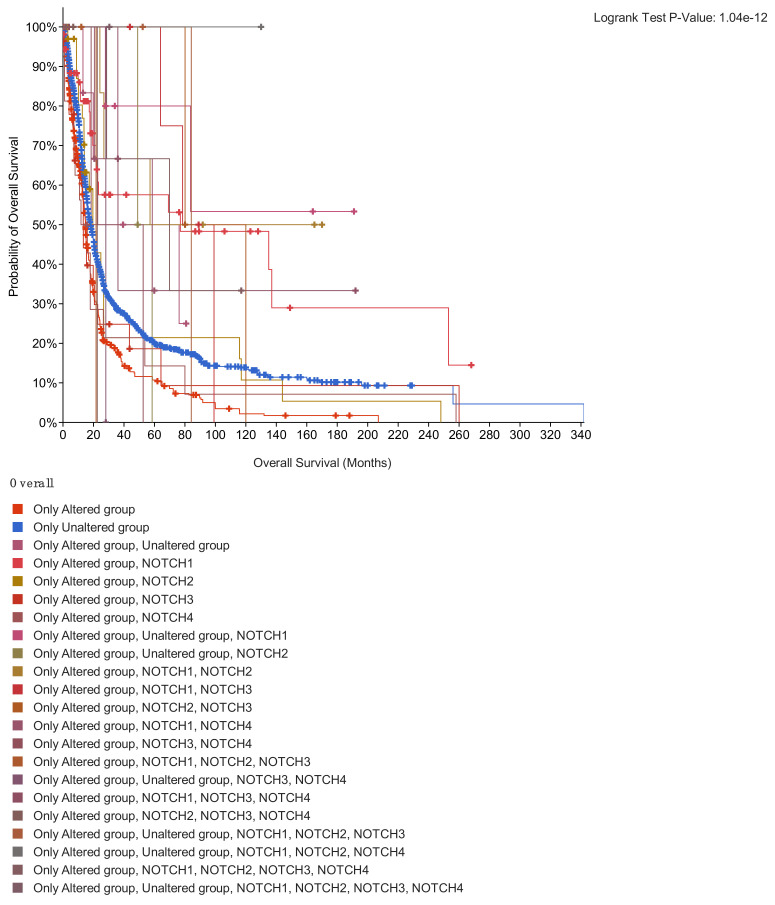
Combined study on NOTCH gene mutations in glioma and GBM and patient’s survival (data obtained from cBioPortal). The survival was lower in patients with NOTCH4 mutations (11 months) and NOTCH3 (13 months) compared to NOTCH1 (77 months) and NOTCH2 (19 months).

**Figure 6 jpm-12-01969-f006:**
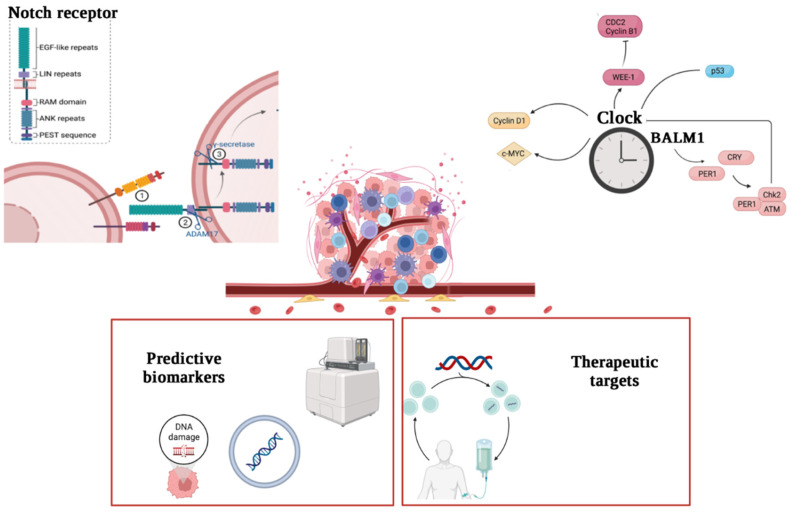
Future diagnostic and therapeutic perspectives in glioma. The analysis of the NOTCH and CLOCK pathways could lead to the identification of predictive markers and therapeutic targets for brain cancer, contributing to the development of personalized medicine.

## Data Availability

Not applicable.

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
