# Peer review of "Overview on Common Genes Involved in the Onset of Glioma and on the Role of Migraine as Risk Factor: Predictive Biomarkers or Therapeutic Targets?"

_jpm, 2022, doi:10.3390/jpm12121969_

Round 1
Reviewer 1 Report
The authors reviewed the role of genes CLOCK and BMLA1 participating in the circadian clock rhythms and NOTCH gene that is significantly correlated to CLOCK/BMLA1 in early diagnosis of the tumor onset and personalized treatment in glioma patients with a simple headache. The manuscript includes interesting contents, but there are some points to be required for correction as following comments.
#1. Authors described that “the significant correlation between CLOCK, BMLA1 and NOTCH genes” (Page 1, Line32, in Abstract), but both the data and references were not presented. The authors should show the evidence of the correlation.
#2. The authors selected a headache as a risk factor for the onset of glioma, but not so many patients with gliomas in both grade I, II and grade III, IV present a headache in the early stage of the diseases (the frequency is 15-21%, Alther B et al, Clin Trans Neurosci, 2020: 1-7). This indicates that a large part of patients with glioma may have not abnormal expression of CLOCK and BMLA1, including NOTCH gene. Are there any differences in the prognosis and response to radio-chemotherapy between patients with headache and without headache?
#3. There are no detail explanation of Figures 2,3 and 4. Please present Figure legends to each Figure.
#4. A part of the sentences in Introduction is redundant. For example, sentences in the first three paragraphs (lines 45-63) and sentences in lines 97-147 should be described more compactly. And it would be better to describe the significance of circadian rhythms more precisely in low-grade and high-grade gliomas.
Author Response
The authors reviewed the role of genes CLOCK and BMLA1 participating in the circadian clock rhythms and NOTCH gene that is significantly correlated to CLOCK/BMLA1 in early diagnosis of the tumor onset and personalized treatment in glioma patients with a simple headache. The manuscript includes interesting contents, but there are some points to be required for correction as following comments.
#1. Authors described that “the significant correlation between CLOCK, BMLA1 and NOTCH genes” (Page 1, Line32, in Abstract), but both the data and references were not presented. The authors should show the evidence of the correlation.
As suggested by reviewer, the authors corrected the sentence, to highlight which the focus of the review was to underline a significant modulation of CLOCK, BMLA1 and NOTCH genes in glioma, particularly praising NOTCH genes family to be good as potentially attractive therapeutic targets for glioblastoma.
#2. The authors selected a headache as a risk factor for the onset of glioma, but not so many patients with gliomas in both grade I, II and grade III, IV present a headache in the early stage of the diseases (the frequency is 15-21%, Alther B et al, Clin Trans Neurosci, 2020: 1-7). This indicates that a large part of patients with glioma may have not abnormal expression of CLOCK and BMLA1, including NOTCH gene. Are there any differences in the prognosis and response to radio-chemotherapy between patients with headache and without headache?
Pre-existing headache represents an important risk factor according to logistic regression, suggesting that patients with pre-existing (primary) headache have a greater predisposition to develop secondary headache. Dull headache occurs significantly more often in patients with GBM, and pulsating headache in patients with glioma (Schankin CJ, Ferrari U, Reinisch VM, Birnbaum T, Goldbrunner R, Straube A. Characteristics of brain tumour-associated headache. Cephalalgia. 2007 Aug;27(8):904-11). Particularly, according to the studies performed after the advent of the modern neurodiagnostic techniques, the incidence of every type of headache in populations of patients affected by brain tumor ranges from 32% to 71% (Palmieri A, Valentinis L, Zanchin G. Update on headache and brain tumors. Cephalalgia. 2021 Apr;41(4):431-437). No data regarding differences in the prognosis and response to radio-chemotherapy between patients with headache and without headache are cited in literature.
#3. There are no detail explanation of Figures 2,3 and 4. Please present Figure legends to each Figure.
As suggested by reviewer, the authors inserted detail explanation in figure legends regarding Figures 2, 3 and 4.
#4. A part of the sentences in Introduction is redundant. For example, sentences in the first three paragraphs (lines 45-63) and sentences in lines 97-147 should be described more compactly. And it would be better to describe the significance of circadian rhythms more precisely in low-grade and high-grade gliomas.
As suggested by reviewer, the authors improved the sentences cited in Introduction, to better highlight the focus of the review. Moreover, the authors better described the significance of circadian rhythms more precisely in low-grade and high-grade gliomas.
Reviewer 2 Report
In this review “Overview on common genes involved in the onset of glioma: predictive biomarkers or therapeutic targets?” authors have associated the common genes of migraine and gliomas, as well as explain about the prognostic and/or therapeutic values of them. Even if there is not strong conclusions about the better and exact role of the genes in glioma, this manuscript bring a good overview regard this interesting relations which could be promising for future important studies.
Title: Title should be improved, once the title did not represent the article content inducing the readers to thing that the review is about common genes of glioma onset in general while it is about the genes common of glioma and migraine.
Line 168: The sentence is confusing and must be improved (Research shows that the CLOCK gene plays a major role as an activator of downstream elements in the pathway critical to the generation of circadian rhythms(32-35) and the cause seems to be found in single nucleotide polymorphism (SNP: rs12649507) in the CLOCK gene).
Line 217: I believe that the firs “in” of the sentence should be “is” - The expression of BMAL1 in involved in glioma biology, both when it is upregulated 217 then downregulated. Interestingly, BMAL1 over expression in high-grade glioma patients.
Line 232: Much has been discovered about imune system and canecr relationship, authors could improve the following paragraph and better explain the consequence of abundant microglia recruitment once imune-supressive macrophages could corroborate with tumor progression (As well, demonstrating the capacity of CLOCK to specifically and directly regulate the chemokine such as OLFML3, novel chemokine recruiting immune-suppressive microglia into the tumor microenvironment, which in turn recruits microglia into the GBM, encourages the design of clinical trials targeting OLFML3 in high CLOCK GBM patients and abundant microglia(61)).
In Line 266 authors cite Cadasil desiase without any previously explanation about it.
Figures in general: All figures have small letters that appear out of focus. Moreover figure 3 is cited in the text before figure 2. Figure 3 is first cited in line 283, while figure 2 is first cited in line 302. Moreover, the text that preceds the citeation of figure 3 in line 283 seems to be not related with Figure 3.
Figure 3: The axes of the graphs are the same, is this correct? If the authors' intention was to show the methylation values for each gene, shouldn't they have used a Vulcan chart?
Figure legends: must be improved, as figure 4 legend that did not explain the meaning of percentage values.
Author Response
In this review “Overview on common genes involved in the onset of glioma: predictive biomarkers or therapeutic targets?” authors have associated the common genes of migraine and gliomas, as well as explain about the prognostic and/or therapeutic values of them. Even if there is not strong conclusions about the better and exact role of the genes in glioma, this manuscript bring a good overview regard this interesting relations which could be promising for future important studies.
#1. Title: Title should be improved, once the title did not represent the article content inducing the readers to think that the review is about common genes of glioma onset in general while it is about the genes common of glioma and migraine.
As suggested by reviewer, the authors improved the title, to better highlight the focus of review, although the focus of the review was to underly the common genetic mutations involved in glioma onset, pointing out how migraine can be a risk factor.
#2. Line 168: The sentence is confusing and must be improved (Research shows that the CLOCK gene plays a major role as an activator of downstream elements in the pathway critical to the generation of circadian rhythms(32-35) and the cause seems to be found in single nucleotide polymorphism (SNP: rs12649507) in the CLOCK gene).
As suggested by reviewer, the authors better rewrote the sentence, to improve the meaning of the speech.
#3. Line 217: I believe that the firs “in” of the sentence should be “is” - The expression of BMAL1 in involved in glioma biology, both when it is upregulated 217 then downregulated. Interestingly, BMAL1 over expression in high-grade glioma patients.
As suggested by reviewer, the authors corrected the grammatical errors.
#4. Line 232: Much has been discovered about immune system and cancer relationship, authors could improve the following paragraph and better explain the consequence of abundant microglia recruitment once immune-suppressive macrophages could corroborate with tumor progression (As well, demonstrating the capacity of CLOCK to specifically and directly regulate the chemokine such as OLFML3, novel chemokine recruiting immune-suppressive microglia into the tumor microenvironment, which in turn recruits microglia into the GBM, encourages the design of clinical trials targeting OLFML3 in high CLOCK GBM patients and abundant microglia(61)).
As suggested by reviewer, the authors improved the paragraph, better explaining the consequence of abundant microglia recruitment once immune-suppressive macrophages could corroborate with tumor progression.
#5. In Line 266 authors cite Cadasil disease without any previously explanation about it.
As suggested by reviewer, the authors better rewrote the sentence to highlight the involvement of NOTCH3 genetic mutations in different disease, including CADASIL.
#6. Figures in general: All figures have small letters that appear out of focus. Moreover figure 3 is cited in the text before figure 2. Figure 3 is first cited in line 283, while figure 2 is first cited in line 302. Moreover, the text that proceeds the citation of figure 3 in line 283 seems to be not related with Figure 3.
As suggested by reviewer, the authors improved the quality of all figures, increasing the letters inside. Moreover, the authors have corrected the chronological order in which the figures are cited.
#7. Figure 3: The axes of the graphs are the same, is this correct? If the authors' intention was to show the methylation values for each gene, shouldn't they have used a Vulcan chart?
The axes of the graphs are the same because the authors wanted to highlight the relationship between copy number alterations (Not mutated, Amplification Gain, Diploid, Shallow Deletion) and the expression levels of NOTCH genes.
#8. Figure legends: must be improved, as figure 4 legend that did not explain the meaning of percentage values.
As suggested by reviewer, the authors improved figure legends and better explain the meaning of percentage values in Figure 4.
Reviewer 3 Report
The manuscript is interesting because they are describing potential therapeutic targets and biomarkers for brain tumors, but there are several points that the authors might to attend before publication:
First of all, during the manuscript is not clear if the authors are talking about all gliomas or only the GBM. The authors are talking about gliomas sometimes and then glioblastoma, but it is important to establish if the study is about all gliomas or glioblastomas because there are a lot of differences between gliomas. I suggest specifying or changing the order of some sentences to make it clear. In addition, I suggest ordering all text, I think that there are different concepts and at some points are a little bit disorder.
There is missing acronym GBM in sentence 52 after glioblastoma. The authors should add "glioblastoma (GBM)" because sentence 56 says GBM directly.
There are missing several references in all text, for example, after: "Glioblastomas are the most aggressive brain tumors; current herapies include: surgical resection, followed by radiotherapy plus simultaneous treatment and maintenance with temozolomide (TMZ)" The current therapy is based on Stupp protocol, which is not referenced; "genes alterations related to migraine as potential risk factors for brain tumor onset."; "Although GBM account for only 15% of primary brain cancer, it represents the most malignant form due to its poor prognosis"; "The overexpression of its specific ligand the DLL3 mRNA was observed in the proneural sub-class of GBM, and in IDH1 mutant LGG.". Please, review that all studies are referenced.
Reference 13 is a review, so I suggest adding the original study or studies because I think that there are more studies related to this topic.
There is the missing meaning of the acronym CSCs, in sentence 137. In the field is well known, but I suggest adding cancer stem cells (CSCs). Similarly, the acronym, BMAL1 is in sentence 64 and then, in sentence 157 the extended name, please change it.
What is the meaning of SCN in sentence 158?
Regarding the CLOCK pathway, at what point it will be better to treat the patient? In sentence 248 the authors comment on the study about the treatment of GBM with Bortezomib, but there is no explanation or relation to the CLOCK pathway. I suggest improving the discussion about the treatment of GBM or glioma with different treatments with CLOCK as a target.
Please, review sentence 283 where Figure 3 is cited, but I think that does not correspond to this figure.
I suggest adding another figure with pathways, both to CLOCK and NOTCH pathways to better understand the different genes cited in the text and their implication in the pathway and the treatments that might be used to treat gliomas, according to the review manuscript.
Finally, have the authors studied the difference in the survival between the expression or mutation of genes in the manuscript?
There are figures about the mutation and % of expression in glioma, but no survival information or response to treatment, which is the main objective of the manuscript. I suggest adding this information to improve the manuscript.
In addition, it could be interesting to study the genes in different gliomas, differencing the gliomas with high grade vs low grade, for example.
Author Response
The manuscript is interesting because they are describing potential therapeutic targets and biomarkers for brain tumors, but there are several points that the authors might to attend before publication:
#1. First of all, during the manuscript is not clear if the authors are talking about all gliomas or only the GBM. The authors are talking about gliomas sometimes and then glioblastoma, but it is important to establish if the study is about all gliomas or glioblastomas because there are a lot of differences between gliomas. I suggest specifying or changing the order of some sentences to make it clear. In addition, I suggest ordering all text, I think that there are different concepts and at some points are a little bit disorder.
As suggested by reviewer, the authors better clarified that the focus of the review was to highlight mutations gene involved in glioma, with particularly attention for its role also in GBM. Moreover, the authors performed a revision of all text.
#2. There is missing acronym GBM in sentence 52 after glioblastoma. The authors should add "glioblastoma (GBM)" because sentence 56 says GBM directly.
As suggested by reviewer, the authors corrected the sentence, putting the missing acronym.
#3. There are missing several references in all text, for example, after: "Glioblastomas are the most aggressive brain tumors; current herapies include: surgical resection, followed by radiotherapy plus simultaneous treatment and maintenance with temozolomide (TMZ)" The current therapy is based on Stupp protocol, which is not referenced; "genes alterations related to migraine as potential risk factors for brain tumor onset."; "Although GBM account for only 15% of primary brain cancer, it represents the most malignant form due to its poor prognosis"; "The overexpression of its specific ligand the DLL3 mRNA was observed in the proneural sub-class of GBM, and in IDH1 mutant LGG.". Please, review that all studies are referenced.
As suggested by reviewer, the authors inserted the missing references in all text, also checking which all studies were referenced.
#4. Reference 13 is a review, so I suggest adding the original study or studies because I think that there are more studies related to this topic.
As suggested by reviewer, the authors added original study regarding the topic discussed.
#5. There is the missing meaning of the acronym CSCs, in sentence 137. In the field is well known, but I suggest adding cancer stem cells (CSCs). Similarly, the acronym, BMAL1 is in sentence 64 and then, in sentence 157 the extended name, please change it.
As suggested by reviewer, the authors performed the corrections indicated.
#6. What is the meaning of SCN in sentence 158?
The acronym SCN indicates suprachiasmatic nucleus (SCN) of the hypothalamus. The authors better specified this term in the text.
#7. Regarding the CLOCK pathway, at what point it will be better to treat the patient? In sentence 248 the authors comment on the study about the treatment of GBM with Bortezomib, but there is no explanation or relation to the CLOCK pathway. I suggest improving the discussion about the treatment of GBM or glioma with different treatments with CLOCK as a target.
The authors highlighted the importance of chronotherapy, described as a therapeutical approach based on patients ‘circadian rhythms as novel concepts in glioma therapy. The connection between CLOCK pathway and Bortezomib treatment lies in the point which scientific literature demonstrated the higher efficacy of low‐dose Bortezomib treatment, the proteasome inhibitor in anticancer therapy, when administered in tumor‐bearing animals at night compared to day/night administration (Wagner PM, Prucca CG, Velazquez FN, Sosa Alderete LG, Caputto BL, Guido ME. Temporal regulation of tumor growth in nocturnal mammals: In vivo studies and chemotherapeutical potential. FASEB J. 2021;35(2):e21231). Unfortunately these data represented new pre-clinical experimental research, without patient feedback. In fact, for high‐grade glioma patients, radiotherapy treatment time of the day (RT‐TTD) did not influence progression‐free survival between patients treated in the morning or afternoon. There is still a lack of multi‐center, double‐blinded random control trials to further elaborate the influence of the CLOCK gene in glioma patients and the effect of chronotherapy treatment in glioma is still controversial and needs further large‐population‐based trials for validation. For this reason, the authors better rewrote the discussion about the treatment for glioma/GBM with CLOCK as target.
#8. Please, review sentence 283 where Figure 3 is cited, but I think that does not correspond to this figure.
As suggested by reviewer, the authors corrected the sentence, better indicating the Figure 3 in in the part of the text relating to it.
#9. I suggest adding another figure with pathways, both to CLOCK and NOTCH pathways to better understand the different genes cited in the text and their implication in the pathway and the treatments that might be used to treat gliomas, according to the review manuscript.
As suggested by reviewer the authors added a new Figure 6, to better describe the CLOCK and NOTCH pathways involved in glioma pathogenesis and the treatments applicable.
#10. Finally, have the authors studied the difference in the survival between the expression or mutation of genes in the manuscript?
In literature, the unsupervised clustering of 13 clock gene transcriptomic signatures from The Cancer Genome Atlas showed that a low period (PER) gene expression was associated with the negative prognosis and enrichment of the immune signaling pathways (De La Cruz Minyety J, Shuboni-Mulligan DD, Briceno N, Young D Jr, Gilbert MR, Celiku O, Armstrong TS. Association of Circadian Clock Gene Expression with Glioma Tumor Microenvironment and Patient Survival. Cancers (Basel). 2021 Jun 2;13(11):2756). Regarding NOTCH mutations, they represent a high-risk factor associated with shorter patient survival, suggesting that NOTCH signaling inhibition contributes to increased glioma aggressiveness (Aoki K., Nakamura H., Suzuki H., Matsuo K., Kataoka K., Shimamura T., Motomura K., Ohka F., Shiina S., Yamamoto T., et al. Prognostic relevance of genetic alterations in diffuse lower-grade gliomas. Neuro Oncol. 2018;20:66–77). However, as suggested by reviewer, the authors performed a study regarding the survival compared to mutation of NOTCH genes using cBioportal, showing as the survival resulted to be lower in patients with NOTCH3 and/or NOTCH 4 mutations (Figure 5).
#11. There are figures about the mutation and % of expression in glioma, but no survival information or response to treatment, which is the main objective of the manuscript. I suggest adding this information to improve the manuscript.
As suggested by reviewer, the authors added a new Figure 5 regarding survival information.
#12. In addition, it could be interesting to study the genes in different gliomas, differencing the gliomas with high grade vs low grade, for example.
The authors focus their attention particularly on NOTCH and CLOCK genes families in glioma and GBM. Further studies will be needed to better analyze the role of genes mutations in glioma with high grade vs low grade. the authors thank for the excellent scientific point given by the reviewer.
Round 2
Reviewer 3 Report
The answers to the comments are appropriate. However, there are missing the legend of figure 6 and seems to have low quality. After this correction, I will accept the article to be published.
Author Response
The answers to the comments are appropriate. However, there are missing the legend of figure 6 and seems to have low quality. After this correction, I will accept the article to be published.
As suggested by reviewer, the authors added the legend of Figure 6 and increased the quality of the image.